# Platelets and Their Role in Hemostasis and Thrombosis—From Physiology to Pathophysiology and Therapeutic Implications

**DOI:** 10.3390/ijms232112772

**Published:** 2022-10-23

**Authors:** Alina Scridon

**Affiliations:** Physiology Department, Center for Advanced Medical and Pharmaceutical Research, University of Medicine, Pharmacy, Science and Technology “George Emil Palade” of Târgu Mureș, 540142 Târgu Mureș, Romania; alinascridon@gmail.com

**Keywords:** antiplatelet agents, anticoagulants, hemostasis, platelets, thrombosis

## Abstract

Hemostasis is a physiological process critical for survival. Meanwhile, thrombosis is amongst the leading causes of death worldwide, making antithrombotic therapy one of the most crucial aspects of modern medicine. Although antithrombotic therapy has progressed tremendously over the years, it remains far from ideal, and this is mainly due to the incomplete understanding of the exceptionally complex structural and functional properties of platelets. However, advances in biochemistry, molecular biology, and the advent of ‘omics’ continue to provide crucial information for our understanding of the complex structure and function of platelets, their interactions with the coagulation system, and their role in hemostasis and thrombosis. In this review, we provide a comprehensive view of the complex role that platelets play in hemostasis and thrombosis, and we discuss the major clinical implications of these fundamental blood components, with a focus on hemostatic platelet-related disorders and existing and emerging antithrombotic therapies. We also emphasize a number of questions that remain to be answered, and we identify hotspots for future research.

## 1. Introduction

From their initial description in the 1870’s, it had already become clear that platelets were exceptional cells that will never stop amazing the scientific community. It soon became clear that although platelets circulate in isolation in the blood, they can rapidly form aggregates at the sites of vascular injury [1], contributing to both hemostasis and thrombotic disease. However, at that point, the saga of platelets was only in its infancy. Observations that span more than half a century demonstrate that, besides their classical roles in clot formation, platelets play extremely versatile functions in many areas of physiology. Platelets’ *α*-granules express receptors that facilitate adhesion with other vascular cells and release a broad variety of mediators that participate in and regulate functions such as chemotaxis, stem cell homing, cell migration, proliferation and differentiation, inflammation, angio- and lymphangiogenesis, the maintenance of lymphatic and blood systems as separate entities, the deposition of matrix proteins, host defense, viral replication, the transport of information, vasomotor function, and many others. Via this plethora of functions, platelets’ *α*-granules contribute to a wide range of physiologic and pathologic processes [2]. Activated platelets release a vast array of growth and angiogenic factors, cytokines, and chemokines (Table 1). Once released, these molecules act to regulate the chemotaxis, inflammation, and vasomotor function critical for restoring the integrity of injured vascular walls, for angiogenesis, and for the growth of new blood vessels in injured tissue areas. These features provide platelets with tremendous potential for wound healing and tissue regeneration, as already demonstrated in settings such as diabetic ulcers, bone or tendon defects, maxillofacial and dental surgery, and corneal diseases [3]. It has also been suggested that platelets could promote rejuvenation and the reversal of aging by acting as a ‘fountain of youth’ [4], although many of these applications are not yet supported by adequate evidence from clinical trials.

However, in parallel, activated platelets also cause alterations in endothelial and white blood cells and release inflammatory mediators, thereby promoting atherosclerosis and atherothrombosis, vascular intima proliferation and restenosis post-angioplasty, and tumor growth, metastasis, and immune evasion, as well as amplifying inflammatory and infectious states [2]. Studies have also shown that platelets release bioactive, antimicrobial peptides and kinocidins and exhibit antimicrobial host defense properties, possessing the unambiguous structural and functional characteristics of immune cells [5]. Platelets thus play central roles as part of the intravascular innate immune system and coordinate adaptive antimicrobial host defense, thereby bridging anti-infectious innate and adaptive immunity. There is evidence that platelets play complex roles in malarial infections, dengue fever, sepsis, and rheumatic diseases, and deficiencies in platelet quantity or quality are increasingly recognized as correlates of infection risk and severity [6]. In addition, complex interactions have been shown to exist between platelets’ inflammatory and hemostatic functions. A prothrombotic platelet phenotype (involving platelet activation and platelet consumption leading to thrombocytopenia) can be seen in conditions with genetic deficiency in complement regulation such as atypical hemolytic uremic syndrome or paroxysmal nocturnal hemoglobinuria [7].

Platelets, therefore, display a plethora of multifaceted functions. However, the main role of platelets remains maintaining normal hemostasis, in conjunction with the coagulation system. The classical theory of hemostasis (Figure 1A) describes a three-step process during which: (1) immediately after vascular injury, the injured vessel undergoes vasoconstriction to limit blood loss at the site of the injury; (2) platelets adhere to the injured vessel wall, activate, and form aggregates, i.e., the platelet plug; which (3) is eventually stabilized by a dense fibrin mesh formed via the coagulation cascade. In reality, however, this process is much more complex. The three phases that ensure hemostasis are by no means independent and their activation is not precisely sequential. Rather, the three main processes that entail normal hemostasis activate simultaneously and continuously potentiate one another throughout the hemostatic process (Figure 1B). Particularly, the contribution of platelets to hemostasis far exceeds their simple participation in ‘primary hemostasis’, and the multiple interactions that exist between platelets, the vessel wall, and the coagulation system complicate the fundamental process of hemostasis.

In this review, we aim to provide a comprehensive view of the complex role that platelets play in hemostasis and thrombosis, and to discuss the major clinical implications of these fundamental blood components, with a focus on hemostatic platelet-related disorders and existing and emerging antithrombotic therapies. We also emphasize a number of questions that remain to be answered, and we identify hotspots for future research.

## 2. Platelet Shape and Structure at Rest and during Activation

Platelets are small (≈2–4 μm), short-living (≈8–10 days), anucleated cell fragments derived from the megakaryocyte lineage. Most platelets circulate in the bloodstream in isolation in a resting, discoid form, without interacting with the vessel wall, but continuously monitoring their surrounding environment via a wide array of receptors and adhesion molecules and are ultimately cleared from the blood at the end of their lifespan. Therefore, continuous platelet production is required to maintain normal platelet counts (i.e., 150–400 × 10^9^ /L in a healthy adult).

In response to vessel injury, platelets activate and rapidly reveal their highly dynamic nature: they undergo massive shape and ultrastructure changes, including the ‘ruffling’ of their plasma membranes due to the emergence of cytoplasmic projections, causing them to take an ‘amoeboid form’. In parallel, platelets undergo granule centralization and discharge. These critical ultrastructural changes mediated by the platelet cytoskeleton allow platelets to adhere to the site of vessel damage, to spread over the injured area, release the content of intracytoplasmic granules, and aggregate with other activated platelets to form the platelet plug, while also promoting fibrin formation and vessel wall repair. The constituents critical for hemostasis are located both on the surface of the platelet membrane and in the cytoplasm, particularly within the granules (Figure 2).

The outer surface of resting circulating platelets is covered by a layer of glycolipid and glycoprotein molecules that form a prominent **glycocalyx**. With its negative net electric charge, the glycocalyx provides a repulsive surface that prevents spontaneous platelet aggregation with other platelets or other blood or endothelial cells [8], while also playing a role in calcium-signaling regulation [9] and in platelet turnover [10].

A wide variety of glycoproteins are embedded in the **platelet membrane**, acting as receptors for various soluble (i.e., platelet activators) and fixed (i.e., platelet adhesion molecules) ligands that mediate platelets’ adhesion to the vessel wall, activation, spreading, and aggregation [11]. Although the platelet plasma membrane does not differ considerably from those of other cells, it does possess some critical features, including the presence of a plasma membrane-based open canalicular system (i.e., a network of membrane invaginations that penetrate the platelet interior) connected with the extracellular space through a multitude of small pores that provides platelets with a membrane surface area much larger than that expected for such small cells (Figure 2). A second platelet canalicular system (i.e., the dense tubular system), derived from the smooth endoplasmic reticulum of megakaryocytes, is not connected with the extracellular fluid, and serves as a store for calcium and various enzymes involved in platelet activation [11].

The platelet plasma membrane contains numerous adhesion and signaling integrin molecules, leucine-rich glycoproteins, immunoreceptors, prostanoids, and G protein-coupled receptors (Table 2).

**Adhesion molecules** expressed on the platelets’ surface are critical for platelets’ adhesion to the vessel wall and to platelets’ interactions with other cells such as leukocytes and endothelial cells [11]. Many of these molecules are expressed in small amounts in the membrane of resting, unstimulated platelets, and their expression increases rapidly upon platelet activation via the fusion of granule membranes, which also express these molecules, with the platelet membrane. Such molecules include P-selectin, glycoprotein (GP) Ib-IX-V, the main platelet receptor for von Willebrand factor (vWF), and GPIIb/IIIa, critical for platelet aggregation by binding vWF, fibronectin, vitronectin, and, particularly, fibrinogen. Platelet **c****ollagen receptors** are essential for platelets’ adhesion to the vessel wall (Table 2). They are mainly represented by GPVI, a member of the immunoglobulin superfamily, and by *α*/*β* integrin, the primary collagen receptor, with critical hemostatic roles. Finally, the **G-protein-coupled receptors** (Table 2) ensure platelet activation through several ligands such as thrombin, adenosine diphosphate (ADP), adenosine triphosphate (ATP), and prostanoids. Thrombin, a key constituent of the coagulation cascade, is among the most powerful platelet activators. Platelet activation by thrombin is mediated by protease-activated receptors (PARs) expressed in the platelet plasma membrane. The activation of PARs occurs via proteolytic cleavage by thrombin and specific ligand unmasking [12]. Thrombin and various synthetic compounds known as thrombin-receptor agonist peptides (TRAPs) bind to and activate PAR-1 and PAR-4, leading to platelet activation, shape change, granule release, and aggregation. However, platelet activation by TRAPs does not involve receptor cleavage. While PAR-1 activation can occur at low thrombin concentrations, higher agonist concentrations are required for the activation of PAR-4. Contrary to human platelets, rodent platelets do not express PAR-1 [13]. Meanwhile, ADP and ATP are known to evoke platelet responses different than those induced by thrombin. Low ADP concentrations have been shown to initiate reversible platelet aggregation, with no granule release, whereas higher ADP concentrations trigger granule release and prostaglandin synthesis, leading to a typical, biphasic, irreversible aggregation process [1]. Platelets’ response to ADP is mediated by P2Y (P2Y1 and P2Y12) G protein-coupled receptors. Although less potent than ADP, ATP can also trigger, via P2 × 1 receptors, platelet activation, shape change, granule release, and can increase platelets’ sensitivity to other agonists (e.g., collagen) [14]. Platelet activation by thromboxane A2 (TxA2), a product of arachidonic acid metabolism under the effect of cyclooxygenase (COX) 1, occurs via TxA2 receptors *α* and *β* activation, with TxA2 receptor *α* being the predominant isoform in human platelets. The activation of these receptors is rapidly followed by platelet activation, shape change, degranulation, aggregation, and increased sensitivity to other agonists, leading to the amplification of the platelet activation response (Figure 3). Receptors for other prostanoids, such as prostaglandin E and prostacyclin, a platelet aggregation inhibitor, are also expressed on the platelet plasma membrane.

Meanwhile, numerous intracytoplasmic **platelet granules** (Figure 2) serve as secretory vesicles capable of releasing their content into the extracellular fluid, and also as ‘guides’ that direct molecules to the plasma membrane during exocytosis. Platelet degranulation is a complex process that involves granules’ merging and fusion in the platelet center, as well as their fusion with the open canalicular system and with the plasma membrane. Three main types of granules (i.e., *alpha*, dense [*δ*], and lysosomal granules) are present in the cytoplasm of quiescent, non-activated platelets, each displaying specific content, ultrastructure, function, and exocytosis kinetics, and each releasing their content in response to different agonists and at different degrees of stimulation (e.g., *α*- and *δ*-granule release occurs in response to low levels of thrombin or ADP, whereas lysosomes’ degranulation requires much higher concentrations of these agonists).

***Alpha* granules** are the largest (≈200–500 nm), most abundant (≈40–80/platelet), and heterogeneous platelet granules [15], being responsible for the granulated platelet cytoplasm’s appearance on peripheral blood smears stained with Romanowsky-type dyes. The vast majority of platelet factors involved in hemostasis—including *β-*thromboglobulins, platelet factor 4, thrombospondin, and P-selectin; numerous coagulation, anticoagulation, fibrinolytic, and antifibrinolytic factors; and a number of molecules involved in platelets’ adhesion to vessel walls such as fibronectin, laminin, and vitronectin—are contained by these granules (Table 1). Factors involved in inflammation, cell growth, and host defense, such as cytokines, chemokines, and growth factors, as well as microbicidal proteins and immune mediators, are also contained in the *α*-granules (Figure 3), which also display a series of membrane-bound receptors such as *α*IIb*β*3, GPVI, the GPIb-IX-V complex, and P-selectin. Hence, the content of *α*-granules pertains to two main functions: hemostasis and immunity. Proteomic studies have identified over 300 soluble proteins released by *α*-granules [16] and it has been postulated that *α*-granules can release specific molecules in response to different agonists [17], although no conclusive evidence has been provided so far in this regard.

Opposed to the *α*-granules, **dense granules** are the smallest platelet granules (≈150 nm), are less abundant (≈3–8/platelet), and, due to their high calcium and polyphosphate content, appear highly electron dense in osmium-stained and whole-mount electron microscopy [11]. High concentrations of serotonin and adenine nucleotides are also present within the dense granules, as well as serotonin, histamine, cations, small GTP-binding proteins, and adhesion molecules such as GPIb, GPIIb/IIIa, and P-selectin (Table 1). Upon platelet activation, the content of *δ*-granules is released into the extracellular fluid via exocytosis, contributing to platelet recruitment and aggregation (via calcium, polyphosphates, and adenine nucleotides), as well as to local vasoconstriction (via serotonin). The third category of platelet granules, **lysosomes**, have an intermediate size between the *α*- and the *δ*-granules (≈200–250 nm). Their acidic internal environment, with hydrolytic enzymes active against several substrates, including some of the extracellular matrix, contribute to thrombus dissolution and extracellular matrix remodeling [11].

## 3. Role of Platelets in (Primary) Hemostasis

Primary hemostasis can be seen as a three-step sequence of events involving (1) platelets’ adhesion to the vessel wall, (2) platelet activation, and (3) the formation of platelet aggregates, which has as an objective the formation of a blood clot that seals a breach formed in a vessel wall.

Seconds after a vessel injury, stationary (e.g., collagen) and mobile (e.g., thrombin, ADP, and TxA2) platelet agonists accumulate locally. Platelets start to **adhere** to the proteins of the subendothelial matrix via their collagen receptors, with more or less involvement of vWF, depending on the local conditions (i.e., local shear rate of flow). Platelets’ contact with the damaged vessel wall is dependent on a unique membrane receptor complex: GPIb-V-IX (Figure 3). The loss of endothelial integrity, such as that occurring during atherosclerotic plaque erosion or rupture, exposes vWF and the collagen of the subendothelial matrix to contact with the circulating platelets. At low shear rates (i.e., 100–1000/s), such as those typical for the venous system or the interior of the atria, platelets can interact directly with the collagen, laminin, and fibronectin molecules of the extracellular matrix (via GPVI; *α*2*β*1 and *α*6*β*1; and *α*IIb*β*3, *α*V*β*3, and *α*5*β*1, respectively). The Von Willebrand factor’s contribution becomes critical in areas with high shear stress (i.e., 1000–4000/s) such as those seen in the arterial system and particularly in areas with stenotic lesions, where circulating vWF becomes immobilized on the exposed subendothelial collagen, binds to GPIb*α* receptors, and unfolds, exposing multiple binding sites for the GPIb-IX-V complex, thereby facilitating the additional, direct binding of GPVI with the subendothelial collagen (via integrin *α*2*β*1) and fibronectin (via integrin *α*5*β*1). Initial platelet tethering to exposed vWF-collagen complexes can thus remain in place long enough for the platelets to become activated by collagen. This process leads to the formation of a platelet monolayer that will further support the adhesion of additional activated platelets. Although vWF-GPIb-mediated platelets’ adhesion to vessel walls can resist high shear stress, this interaction is transient, making this latter phase a mandatory step for stable platelet adhesion. The interaction between vWF and GPIb*α* facilitates the deceleration of circulating resting platelets and allows them to ‘roll over’ the vessel wall, thereby allowing other platelet receptors to interact with components of the exposed extracellular matrix and/or locally generated soluble agonists such as thrombin. The fibrinogen, fibronectin, vitronectin, and vWF released from the platelet *α*-granules further strengthen platelets’ adhesion to the vessel wall by forming cross-bridges between platelet GPIIb/IIIa receptors and the endothelial *α*V*β*3 integrin or the intercellular adhesion molecule (ICAM) 1 [18]. In turn, integrin *α*/*β*-induced adhesion induces, through phospholipase C-dependent GTPase Rap1b stimulation, platelet GPIIb/IIIa receptors’ activation [19]. Together, these features explain why platelets’ adhesion to the vessel wall is stronger in areas with high local shear stress, why platelets play more important roles in arterial than in venous thrombosis, and why, despite the high flow velocities encountered in areas with vascular stenosis, blood clots generally form precisely in those areas.

The activation of platelet GPVI collagen receptors rapidly triggers phospholipase C activation, the increase of cytosolic calcium, and the hydrolyzation of phosphatidylinositol-4,5-bisphosphate into inositol 1,4,5-trisphosphate and 1,2-diacylglycerol, leading to platelet **activation** (Figure 3). In turn, inositol trisphosphate will mobilize calcium from the intracellular stores, with the consequent additional entry of calcium from the extracellular space, thus leading to the substantial amplification of the platelet calcium signal. In physiological conditions, this prolonged rise in intracellular calcium is most likely triggered by phospholipase C activation and subsequent calcium inflow induced by collagen (via the GPVI receptors) and by high thrombin/vWF concentrations (via PAR-1 and GPIb receptors) [20]. During activation, platelets undergo massive shape and ultrastructural changes and release a series of mediators from their *α*-granules, including TxA2 and ADP. Once released, these mediators will bind to their specific platelet receptors and will act as additional platelet activators via autocrine and paracrine mechanisms, thus amplifying platelet activation and recruiting novel platelets into the hemostatic process (Figure 3). Meanwhile, membrane-bound diacylglycerol, together with calcium, activates protein kinase C, resulting in further integrin activation and platelets’ spreading and degranulation. Adenosine diphosphate-induced platelet activation is initiated by P2Y1 receptors’ activation and is completed and amplified by the activation of the dominant P2Y12 receptors. Once activated, the G_i_-coupled P2Y12 receptors trigger a sequence of intracellular events that lead to the inhibition of platelet adenylate cyclase, with a consequent decrease in cyclic adenosine monophosphate, thereby causing mild negative feedback upon platelet activation via the protein kinase A signaling pathway, coupled with the Akt pathway, and integrin’s activation via Rap1b’s inactivation [14]. The activation of the G_q_-coupled P2Y1 receptors activates the phospholipase C pathway, leading to increased intracellular calcium, the activation of Rap1b, and, ultimately, to GPIIb/IIIa receptors’ activation, which is involved in platelet aggregation (Figure 3). Similar platelet changes are induced by thrombin, which acts as an exogenous platelet agonist (Figure 3). The activation of PAR-1 and PAR-4 thrombin receptors also results in phospholipase C pathway activation, calcium mobilization, and protein kinase C activation, involving fast but reversible PAR-1, followed by sustained PAR-4 cleavage [1], whereas TxA2 receptors’ activation results in phospholipase C and RhoGEF activation, with subsequent platelet degranulation, generation, and the release of lipid mediators, as well as GPIIb/IIIa receptors’ activation [21]. Although extracellular matrix components are crucial for hemostatic mass formation, G protein-coupled receptors and their ligands are key to the hemostatic process. Soluble ligands can activate platelets located in the outer layers of a frowning thrombus, whereas extracellular matrix components cannot. The latter provide, however, a system that delicately adjusts the strength and duration of platelet activation and are capable of triggering extremely fast intracellular signaling pathways necessary for platelets’ adhesion under flow conditions [22].

In addition to the substantial shape changes and degranulation, platelet activation also entails the externalization of anionic phospholipids on the outer surface of the platelet membrane, providing a scaffold for the progress of the coagulation cascade. Resting, quiescent platelets typically expose an anticoagulant membrane due to an asymmetrical distribution of phospholipids, with only phosphatidylcholine and sphingomyelin being exposed on the outer membrane leaflet. Both molecules are electrically neutral and, therefore, cannot bind to clotting proteins with appreciable affinity, whereas the anionic phospholipids phosphatidylserine and phosphatidylethanolamine, responsible for binding to many of the blood coagulation proteins, are sequestered in the inner leaflet of the membrane lipid bilayer facing the cytosol. This arrangement is actively maintained by phospholipid transporters and prevents the membranes of resting, non-activated platelets from supporting coagulation [23]. However, when platelets become activated, the function of the phospholipid transporters is altered, resulting in the ‘scrambling’ of the membrane asymmetry and the transfer of phosphatidylserine and phosphatidylethanolamine on the outer membrane leaflet. The loss of this asymmetry, with anionic phospholipids moving toward the outer membrane’s lipid bilayer, provides a procoagulant surface for the sequential activation of coagulation enzymes, providing the platelet plasma membrane with the critical ability to support the generation of thrombin. Furthermore, the exposure of phosphatidylserine on the outer surface of the platelet membrane prompts the platelets to continue to dramatically change their shape—they lose most of their cytoskeletal structure, rapidly swell, and become balloon-like structures, further increasing their procoagulant surface. Platelets’ inability to expose phosphatidylserine on their outer plasma membrane, as seen in Scott syndrome, significantly diminishes platelets’ procoagulant activity.

Phosphatidylserine’s exposure on the platelet surface is triggered during platelet activation by strong agonists, particularly collagen plus thrombin. The strong exposure induced by these two agonists ensures that the procoagulant transformation of platelets occurs at sites where coagulation is desired (i.e., where the collagen of the vessel wall is exposed) or is already initiated (i.e., where thrombin is present in sufficiently large amounts) [24]. In parallel with the externalization of anionic phospholipids on the platelet plasma membranes, membrane blebs—phosphatidylserine-rich microvesicles—are generated and released into the bloodstream. Platelet microvesicles’ formation occurs via the intervention of calpain, which degrades and dissociates the membrane skeleton from the plasma membrane, facilitating the protrusion of membrane patches, which will no longer interact with the platelet cytoskeleton [25]. Platelet-derived microvesicles contain platelet cytoskeletal proteins and membrane GP Ib, IIb, IIIa, and IV, which contribute to the clot-promoting activity of plasma [1]. Studies suggest that, in addition to their contribution to normal hemostasis, platelet microvesicles also play a role in the thrombotic risk associated with several diseases, particularly via the tissue factor pathway. Circulating platelet-derived microvesicles have been detected in patients with disseminated intravascular coagulation, antiphospholipid syndrome, thrombotic thrombocytopenic purpura, heparin-induced thrombocytopenia, and transient ischemic attacks [1].

Although the sources and the intracellular pathways triggered by the main platelet agonists are different, the consequence of platelet receptors’ activation is common, to some extent, for all platelet agonists: they all lead to the amplification of platelet activation, the recruitment of additional platelets, and to the activation—via inside-out signaling—of the most abundant platelet adhesion molecules, the GPIIb/IIIa receptors, which are critical for platelet **aggregation** (Figure 3). The interaction of this integrin with immobilized ligands is the most critical step in stable hemostatic plug formation, alterations in GPIIb/IIIa activation leading to severe bleeding, and/or thrombotic disorders. In resting platelets, the GPIIb/IIIa receptors are in a closed conformation, with their heads being massively bent over in a compact *V*-shape. Once activated, GPIIb/IIIa receptors undergo important conformational changes. The *α* and *β* domains of the headpieces shift from the closed to an open, high-affinity conformation, exposing the binding sites for their ligands. This shift allows GPIIb/IIIa receptors to pass from a low-affinity to a high-affinity state and to bind their ligands—vWF, fibronectin, vitronectin, and mainly fibrinogen—and thus form bridges with the neighboring activated platelets, forming platelet aggregates (Figure 3). Fibrinogen’s binding to the GPIIb/IIIa receptors leads to receptor grouping, triggers additional platelet activation, and leads (via an outside-in signaling pathway that involves Src family kinases and Syk) to more stable, irreversible platelet aggregation and to clot retraction, limiting the growth of the hemostatic mass and stabilizing the clot structure [26]. In parallel, a number of inhibitory molecules released by the vascular endothelium, such as nitric oxide, prostacyclin, and CD39, prevent unwanted platelet activation and limit the hemostatic process at the site of vascular injury [27]. Furthermore, accumulating evidence indicates that, following the initial stimulation, hemostatic clot formation involves the limitation of platelet activation by providing a restricted environment where agonists can accumulate. This ensures that platelet activation has a heterogeneous, but not random, distribution. While certain platelets become fully activated, change their shape, and degranulate, others are only partially activated, with the more stable core of the clot containing the former, in a densely packed structure, and the less stable shell containing the latter, in a less tightly packed configuration. This platelet subpopulations’ gradient is accurately mirrored by agonist distribution gradients [28]. Moreover, recent data suggest that thrombin activity is restricted to the core region, whereas ADP and TxA2 appear to be the main drivers for platelet accumulation in the shell of the hemostatic clot, explaining the limited impact that the TxA2 and ADP pathway inhibitors exhibit on the clot core [29].

## 4. Clinical Implications of Platelet (Patho)Physiology

### 4.1. Platelet Abnormalities Translate into Clinically Relevant Dysfunctions of (Not Only) Primary Hemostasis

Rare but clinically significant platelet disorders caused by GPIb and GPV (i.e., Bernard-Soulier syndrome) or GPIIb/IIIa (i.e., Glanzman syndrome) deficiencies are associated with bleeding diathesis, whereas platelet granule abnormalities are commonly associated with only mild bleeding. Other genetic platelet abnormalities, such as the naturally occurring Leu/Pro 33 variant of GPIIIa, the presence of the 807T allele of the *α*2 gene, and different polymorphisms in the GPIIb gene, have all been related to an increased susceptibility to arterial thrombosis and myocardial infarction in different clinical studies [30].

However, platelets’ function extends far beyond their contribution to primary hemostasis. Platelets also act as a rich reservoir of coagulation factors, as well as a scaffold mandatory for the progress of the coagulation cascade. Hence, the impact of platelet abnormalities also extends far beyond primary hemostasis, affecting the process of coagulation as well. Tissue factor’s interaction with coagulation factor VII is a critical initial step in the coagulation process. Since tissue factor is an integral membrane protein, tissue factor-activated factor VII interactions require that the tissue factor be expressed on the surface of the cell membrane. Traditionally, it was believed that tissue factor was only expressed by extravascular, subendothelial tissues such as smooth muscle cells, macrophages, and fibroblasts. Studies have shown, however, that tissue factor can also be expressed by endothelial and circulating cells, such as monocytes and platelets [31]. The expression of tissue factor by these latter cells could contribute to the increased risk of disseminated intravascular coagulation in patients with sepsis and to thrombosis in cancer patients. In addition, platelet-derived tissue factor microparticles have been shown to circulate in the blood, and the levels of these microparticles have been shown to be significantly higher in patients with diabetes or atherosclerosis, contributing to the increased risk of thrombosis seen in these populations [32]. The pool of platelet-released factor V also seems to be highly relevant. Residual factor V secretion by activated platelets appears to be sufficient to prevent severe bleeding in patients with congenital factor V deficiency, which present only mild bleeding disorders, whereas platelet factor V deficiency is associated with a bleeding phenotype in patients with Quebec platelet disorder [33]. Similarly, platelet vWF has been shown to partially compensate for the lack of plasma vWF in pigs with severe von Willebrand disease [34]. The critical role of phosphatidylserine exposure in coagulation is exemplified by the bleeding phenotype seen in patients with Scott syndrome, an isolated deficiency in platelets’ procoagulant activity caused by mutations in genes encoding for the phospholipid scramblase. In these patients, although the platelets aggregate normally in response to all agonists, they display a markedly decreased exposure of anionic phospholipids on their surface, have a reduced number of activated factors V and VIII binding sites, and fail to promote prothrombin and factor X activation [35].

Conversely, since thrombin is one of the most powerful platelet activators, coagulation abnormalities will also disturb normal platelet function. Indeed, patients with von Willebrand disease present not only coagulation, but also platelet disorders, as demonstrated by the markedly impaired thrombin generation time measured in the plasma of type 3 von Willebrand disease patients, which lack vWF in plasma, platelets, and endothelial cells, whereas this is not the case when thrombin generation time is measured in the presence of normal platelets [36]. In patients with Owren’s parahemophilia (with congenital factor V deficiency), the existence of functional platelet-derived coagulation factor V is thought to support enough thrombin generation to prevent severe bleeding in patients with virtually undetectable plasma factor V [33]. Meanwhile, the increased tendency for bleeding in patients with Bernard–Soulier syndrome, which lack functional GPIb and GPV receptors, appears to be due to decreased platelet activation in response to thrombin [37].

### 4.2. The Functional Properties of Platelets Explain Their Stronger Implications in Arterial Than Venous Thromboses

All thrombi, regardless of their location, contain platelet aggregates, fibrin, and trapped red blood cells. However, the proportions of these elements are considerably different in the arterial and the venous territories. Arterial thrombosis, such as that occurring in myocardial infarctions, acute limb ischemia, or strokes, usually occurs as a result of an atherosclerotic plaque rupture in vascular areas with high wall shear stress and is generally characterized by obstructive, ‘white thrombi’ that are rich in platelets disposed in large aggregates, have a modest fibrin content, and relatively few trapped red blood cells (Figure 4A). Meanwhile, venous thrombosis occurs in areas with low shear stress and is generally characterized by ‘redder’ thrombi, with more trapped red blood cells and a higher fibrin and lower platelet content, the latter being mainly recruited as single cells (Figure 4B) [27]. Consequently, arterial thrombosis has generally been regarded as a platelet- and venous thrombosis as a coagulation-related disease. In line with this concept, anticoagulant agents have been shown to efficiently prevent venous thrombosis [38], whereas antiplatelet agents have been shown to be highly efficient for arterial thrombosis therapy and secondary prevention [39].

The more important role played by the platelets in the arterial than in the venous territory is mainly related to the high shear stress characteristic of the arterial circulation. In the arteries, and particularly in stenotic areas, where shear stress can reach values as high as 10,000/s, vWF-mediated platelets’ adhesion to the vessel wall via the GPIb-V-IX complex increases in parallel with the wall shear rate, and hence with the flow rate. The same conditions exhibit opposite effects on the coagulation system, washing out coagulation factors and impeding thrombin accumulation, thus limiting coagulation cascade efficacy [40]. These mechanisms clearly explain why platelets play more important roles in the arterial circulation, whereas coagulation factors are more important in the venous system.

In reality, however, this view of thrombosis oversimplifies a very complex process that involves blood stasis, the activation of the vascular endothelium, innate immunity, platelets, and coagulation factors. There is strong evidence for systemic platelet activation in patients with acute venous thromboembolism [41]. In the same vein, platelets have been shown to play critical roles in the initiation and propagation of venous thrombi, and platelet inhibition or depletion suppressed thrombus formation in mouse models of venous thrombosis [42]. Furthermore, studies have shown aspirin and clopidogrel to be capable of attenuating the risk of venous thrombosis in murine venous stasis models [43], as well as in patients undergoing orthopedic surgery [44], and to reduce the risk of deep vein thrombosis recurrence by 32% compared with a placebo, without a significant increase in the risk of bleeding [45], although their effectiveness in venous thrombosis is clearly inferior to that of the oral anticoagulants [46]. Similar results were also reported for patients with atrial fibrillation [47,48]. The ongoing large pragmatic Comparative Effectiveness of Pulmonary Embolism Prevention after Hip and Knee Replacement (PEPPER) trial (NCT02810704)—designed to evaluate the effects of three upfront antithrombotic strategies (i.e., aspirin 81 mg BID, warfarin dose-adjusted to achieve a target international normalized ratio of 2.0, and rivaroxaban 10 mg once daily) on the composite of all-cause mortality and symptomatic venous thromboembolism [49]—is expected to yield more data regarding aspirin’s efficacy in primary venous thromboembolism prevention.

In parallel, while thrombin clearly plays a critical role in venous thrombosis, this procoagulant enzyme is also the most potent physiological platelet activator, which places thrombin as a key element for not only venous but also arterial thrombosis. Indeed, thrombin has been shown to initiate arterial thrombosis and essentially all experimental thrombosis models have been shown to be sensitive to deficiencies in either platelets or coagulation factors [50]. Furthermore, in the landmark Cardiovascular Outcomes for Peoples Using Anticoagulant Strategies (COMPASS) trial, a combination of a low dose of the activated factor X inhibitor rivaroxaban and the antiplatelet drug aspirin was more effective than aspirin alone for the prevention of stroke, myocardial infarction, and cardiovascular death in patients with stable coronary or peripheral artery disease [51]. Similarly, in the Anti-Xa Therapy to Lower cardiovascular events in addition to Aspirin with or without thienopyridine therapy in Subjects with Acute Coronary Syndrome–Thrombolysis in Myocardial Infarction (ATLAS ACS 2-TIMI 51) study, the combined inhibition of platelets (with aspirin and a P2Y12 antagonist) and coagulation (with rivaroxaban) reduced the risk of in-stent thrombosis compared to dual antiplatelet therapy alone [52]. Early clinical trials also demonstrated a clinical benefit of vitamin K antagonists in arterial thrombosis, although at the expense of increased bleeding [53]. Moreover, studies have shown that in patients with acute myocardial infarction, the platelet and fibrin content of coronary thrombi varies, with fibrin becoming more abundant, and platelet content decreasing in direct relationship with the duration of the ischemia [27], observations that may have critical implications on the optimal timing of antithrombotic therapy administration.

### 4.3. Antiplatelet (and Anticoagulant) Drugs Target Key Mechanisms of Platelets’ Adhesion, Activation, and Aggregation

Hemostasis is a physiological process critical for survival. Meanwhile, thrombosis is amongst the leading causes of death worldwide, making antithrombotic therapy one of the most crucial aspects of modern medicine. This context explains the broad range of antiplatelet agents that clinicians have at their disposal. Four main classes of drugs (i.e., COX-1, P2Y12, PAR-1, and *α*IIb*β*3 inhibitors) are currently in use, alone or in various combinations, to counteract platelet hyperreactivity and arterial thrombosis (Figure 3). Several other drug classes, such as activated *α*IIb*β*3, *α*IIb*β*3 outside-in signaling, novel PAR (e.g., parmodulins, pepducins, and PAR-4 inhibitors), phosphatidylinositol 3-kinase-*β*, and protein disulfide-isomerase inhibitors are currently under evaluation with the promise of providing a safer inhibition of thrombosis, with minimal perturbations of hemostasis [27]. A detailed description of these existing and emerging drugs is beyond the scope of the present paper.

Importantly, however, given the complex interactions that exist between platelets and the coagulation system, anticoagulant agents can also exhibit antiplatelet effects. The dual antiplatelet and anticoagulant effect of anticoagulant agents is particularly important in light of the contemporary trend toward combining direct oral anticoagulants with antiplatelet therapy, which could affect the risk of bleeding via a synergistic effect [54] and potentially influence the choice of treatment based on the patient’s risk profile, particularly if potent high-dose anticoagulants and/or potent antiplatelet agents (such as PAR-1 antagonists) are considered for use. The same synergy could also explain, however, the increased efficiency obtained by combining COX-1 with activated factor X inhibitors compared with COX-1 inhibition alone in patients with stable atherosclerotic vascular disease [51]. Clinical studies have also shown that the direct thrombin inhibitor dabigatran decreases platelet aggregation, while also enhancing platelet-mediated fibrinolysis in patients with atrial fibrillation [55,56,57]. However, given that most clinical studies indicate a mild numerical increase, and not a decrease, in the frequency of myocardial infarction following direct thrombin inhibition [58,59], the clinical significance of these findings remains unclear. Moreover, several studies have associated dabigatran administration with an increase in platelet reactivity and this has been at least partly attributed to altered GPIb*α*-thrombin interaction in the presence of shear forces [60] as well as increased PAR-1 [61] and PAR-4 [62] expression on the platelets’ surface following long-term thrombin inhibition [63]. This finding could be of particular importance since PARs can also be activated by serine proteases other than thrombin, including plasmin; activated protein C; matrix metalloproteases 1, 2, and 13; elastase; proteinase-3; granzyme; neutrophil-derived cathepsin G; or calpain [64].

Recent studies indicate that, in addition to thrombin, activated factor X can also act, via PARs, as a direct platelet activator, suggesting that activated factor X inhibition could affect platelet function via a thrombin-independent mechanism [64]. However, the data regarding the effects of direct activated factor X inhibitors on platelet function remain unclear. In some studies, rivaroxaban did not appear to affect platelet aggregation in response to ADP, collagen, TxA2, or thrombin [65]; in other studies, rivaroxaban exerted strong antiplatelet effects [66,67], whereas in others still rivaroxaban was associated with an increase in platelet aggregation [68]. In experimental settings, the direct inhibition of activated factor X using high but not low concentrations of apixaban has also been shown to reduce platelet aggregation as well as fibrin generation [69]. More importantly, studies suggest that apixaban at doses one fifth to one half of that recommended for stroke prevention in atrial fibrillation patients could reduce the formation of large platelet aggregates, while still allowing fibrin formation, thus preserving its contribution to hemostasis [69]. Data concerning the effects of vitamin K antagonists on platelet function are also controversial, with some studies showing that platelet aggregation is not affected by warfarin [70], whereas others showed reduced platelet aggregation following vitamin K antagonists’ administration [71]. Meanwhile, negatively charged activated factor X inhibitors have been shown to not exhibit inhibitory platelet effects. On the contrary, unfractionated heparin and fondaparinux are known to increase platelet functions via non-immune mediated mechanisms [72,73].

## 5. Gaps in Knowledge and Future Research

Advances in biochemistry, molecular biology, and the advent of ‘omics’ techniques have provided crucial data for our understanding regarding the complex structure and functions of platelets and their interactions with the coagulation system. Next-generation sequencing, RNA-sequencing, and platelet proteomics are expected to further unravel the complex processes involved in platelets’ adhesion, activation, and aggregation; in signal transduction, granule secretion, and platelet cytoskeletal changes; and to provide valuable pharmacogenetic information. Advancements in laboratory testing could also provide the means to accurately evaluate platelet function and predict clinical events, similarly to what has already been achieved in other clinical settings [74,75].

Drugs that interfere with platelet function are under continuous development. Antagonists of PAR-1 have already been approved for clinical use and PAR-4 antagonists showed great promise in preclinical studies [76]. Other therapies are also expected to emerge in the near future. The next frontier in this area will probably be reached when antithrombotic drugs that manage to differentiate thrombosis from hemostasis and target them discriminately make their way into clinical practice, thereby offering the prospect of a safer antiplatelet therapy.

The exact short- and long-term effects of the oral anticoagulants on platelet function is another open area for research. This is particularly important given the expanding use of direct oral anticoagulants, especially in patients exposed to such agents for decades, such as those with atrial fibrillation [74,77]. If long-term direct thrombin and/or activated factor X inhibition induce a clinically-relevant increase in platelet reactivity, this effect may translate into an increased risk of thrombosis in certain high-risk populations [78,79,80], particularly following abrupt drug discontinuation [81,82], or could potentially interfere with patients’ responses to concomitant antiplatelet therapy [83,84].

## 6. Conclusions

With their exceptional structural and functional features, platelets play critical roles in hemostasis, vasomotor function, and immunity. In hemostasis, their contribution far exceeds their simple participation in the formation of the ‘platelet plug’. The multiple interactions that exist between platelets, the vessel wall, and the coagulation system complicate the fundamental process of hemostasis and particularly that of thrombosis. Antithrombotic drugs that interfere with platelet functions are under continuous development. The next frontier in this area will probably be reached when antithrombotic drugs that manage to differentiate thrombosis from hemostasis make their way into clinical practice, offering the prospect of safer antiplatelet therapy. Given the rapidly expanding use of the direct oral anticoagulants, establishing their exact impact on platelet functions is another open area for research.

## Figures and Tables

**Figure 1 ijms-23-12772-f001:**
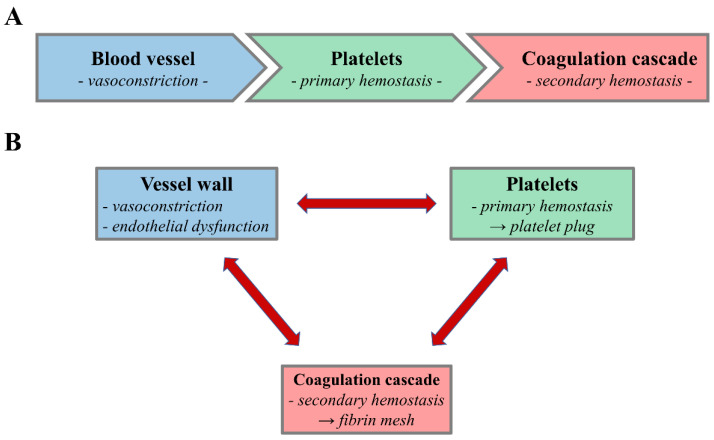
Schematic image of the hemostatic process. (**A**) Classic view of hemostasis as a three-step process involving vasoconstriction and primary and secondary hemostasis as independent and sequential events. (**B**) Current view of hemostasis as a complex process in which the vessel wall, the platelets, and the coagulation system collaborate and continuously influence one another.

**Figure 2 ijms-23-12772-f002:**
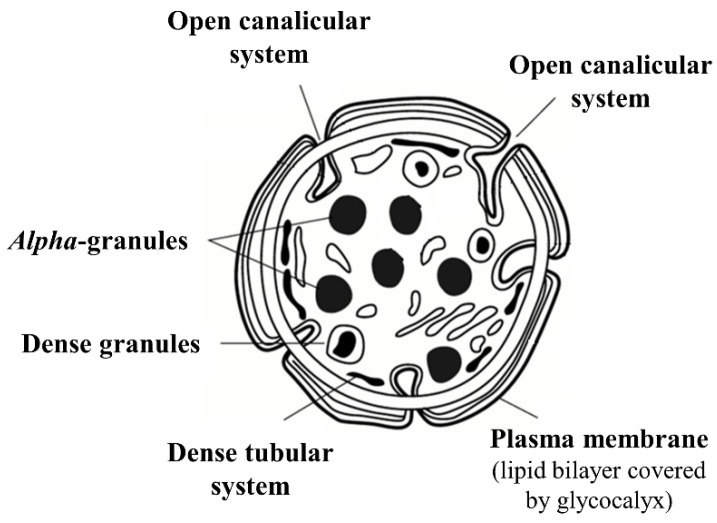
Schematic image of platelets. Resting platelets have asymmetrical distribution of phospholipids: the anionic phospholipids phosphatidylserine and phosphatidylethanolamine, responsible for binding many of the blood coagulation proteins, are sequestered in the inner leaflet of the membrane lipid bilayer facing the cytosol, whereas the electrically neutral phosphatidylcholine and sphingomyelin are exposed on the outer membrane leaflet. This arrangement prevents the membranes of resting platelets from supporting coagulation. When platelets become activated, the function of the phospholipid transporters is altered, leading to transfer of anionic phospholipids on the outer membrane leaflet. Loss of this asymmetry provides a procoagulant surface for sequential activation of coagulation enzymes. The outer surface of resting circulating platelets is covered by a prominent glycocalyx that prevents spontaneous platelet aggregation. Platelets possess a plasma membrane-based open canalicular system connected with the extracellular space through a multitude of small pores that increases the platelet membrane’s surface area. A second platelet canalicular system that is not connected to the platelet’s exterior—the dense tubular system—serves as a store for calcium and for various enzymes involved in platelet activation. Platelet alpha and dense granules contain a large number of substances critical for hemostasis, as well as for vasomotor function and immunity.

**Figure 3 ijms-23-12772-f003:**
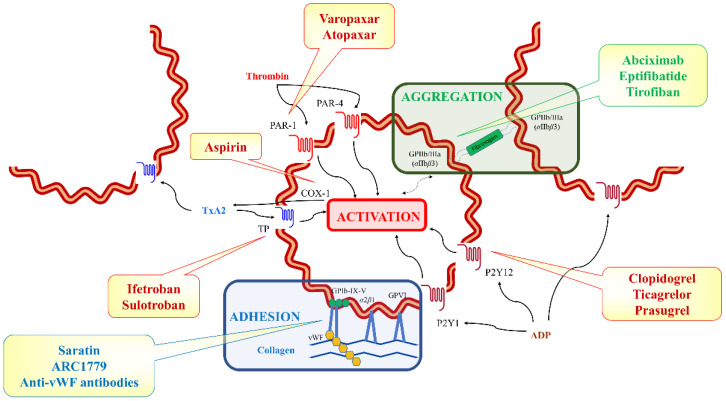
Main receptors and ligands involved in platelets’ adhesion, activation, and aggregation. The boxes indicate some examples of existing or emerging antiplatelet therapies. ADP—adenosine diphosphate; COX-1—cyclooxygenase-1; GP—glycoprotein; PAR—protease-activated receptors; TP—thromboxane receptor; TxA2—thromboxane A2; vWF—von Willebrand factor.

**Figure 4 ijms-23-12772-f004:**
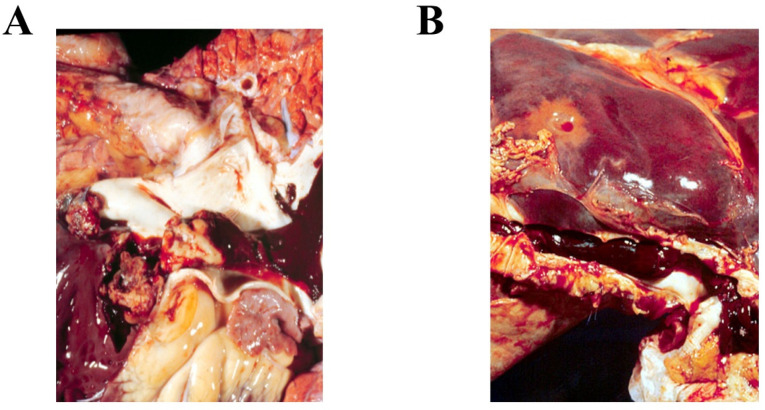
Macroscopic appearance of a (**A**) white thrombus in a patient with carotid artery occlusion and of a (**B**) red thrombus in a patient with deep vein thrombosis.

**Table 1 ijms-23-12772-t001:** Main classes of substances present in platelet granules and cytosol.

***Alpha*-Granules**
*Growth and angiogenic factors* (e.g., platelet-derived growth factor, fibroblast growth factor, vascular endothelial growth factor, connective tissue growth factor, epidermal growth factor, transforming growth factor *β*, insulin-like growth factor 1)
*Cytokines and chemokines* (e.g., interleukin-1*β*, CD40 ligand, CCL2, CCL3, CCL5, CXCL1, CXCL4, CXCL12, CXCL16, platelet factor 4)
*Adhesion molecules* (e.g., P-selectin, fibronectin, vitronectin, von Willebrand factor)
*Coagulation factors* (e.g., factors V, XIII, von Willebrand factor, high-molecular-weight kininogen, fibrinogen)
*Anticoagulation factors* (e.g., tissue factor pathway inhibitor, protein S)
*Fibrinolytic factors* (e.g., plasmin, plasminogen)
*Antifibrinolytic factors* (e.g., *α*2-antiplasmin, thrombin activatable fibrinolysis inhibitor, plasminogen activator inhibitor-1)
*Other molecules* (e.g., albumin, calcitonin, angiotensinogen, thrombospondin)
**Dense granules**
Serotonin Histamine Adenine nucleotides (ADP, ATP) Cations (e.g., calcium, magnesium), polyphosphate Adhesion molecules (e.g., P-selectin)
**Lysosomal granules**
Cathepsin D, E Carboxypeptidase A, B Acid phosphatase Arylsulfatase
**Cytosol**
*Adhesion molecules* (e.g., P-selectin, fibronectin, vitronectin, fibrinogen, thrombospondin, von Willebrand factor)
*Coagulation factors* (e.g., factors V, von Willebrand factor)
*Platelet activators* (e.g., platelet-activating factor, TxA2)
*Vasoconstrictors* (e.g., TxA2, 12-hydroxyeicosatetraenoic acid)

ADP—adenosine diphosphate; ATP—adenosine triphosphate; GP—glycoprotein; TxA2—thromboxane A2.

**Table 2 ijms-23-12772-t002:** Platelet receptors critical for the hemostatic function.

Function in Hemostasis	Receptors	Receptor Family	Main Ligands
Platelet adhesion to the vessel wall	GPIb-IX-V *	Leucine-rich repeat	vWF, thrombospondin-1, thrombin, factors XI, XII, P-selectin
	GPVI	Immunoreceptors	Collagen, laminin
	*α*2*β*1	Integrins	Collagen, laminin
	*α*6*β*1		Laminin
	*α*5*β*1		Fibronectin
	*α*V*β*3		Vitronectin, vWF, fibronectin, fibrinogen
	*α*IIb*β*3 (i.e., GPIIb/IIIa) **		Fibrinogen
Platelet activation	PAR-1, PAR-4	G protein-coupled receptors	Thrombin
	P2Y1, P2Y12		ADP
	TP*α*, TP*β*		TxA2
	PGE_2_ receptor		PGE_2_
	5-HT2A		Serotonin
	P2 × 1	Ion channel	ATP
Platelet aggregation	Activated *α*IIb*β*3 (i.e., GPIIb/IIIa)	Integrins	Fibrin, vWF, thrombospondin-1, fibronectin

* —affected in Bernard–Soulier syndrome; ** —affected in Glanzman thrombasthenia. ADP—adenosine diphosphate; ATP—adenosine triphosphate; GP—glycoprotein; HT—hydroxytryptamine; PAR—protease-activated receptor; PG—prostaglandin; TP—thromboxane receptor; TxA2—thromboxane A2; vWF—von Willebrand factor

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
