# Peer review of "Platelets and Their Role in Hemostasis and Thrombosis—From Physiology to Pathophysiology and Therapeutic Implications"

_ijms, 2022, doi:10.3390/ijms232112772_

Round 1

Reviewer 1 Report

This review article is very well written and easy to follow. I have some minor  suggestions:

Page 1, line 26: half a century

Page 5: first and third paragraphs have very similar data. I think the authors can combine this two parts.

Page 6, line 165: I believe the authors would like to refere the reader to Table 2.

Page 8, line 230: the authors refere the reader to Figure 2, but I could not find P-selectin by its name in that figure.

Author Response

Please find our answers to your comments in the attached file.

Reviewer 2 Report

The reviews by Alina Scridon is very well written and easy to read. Although  general platelet function chapters reflected common knowledge, it is clear and will interest a large scale of reader.

I noticed one important mistake that should be corrected. P7 L176, author write that PAR3 is not express in rodent platelets, while it is in fact PAR1 that is not observed.

Author Response

(The authors gave the same response as above.)

Reviewer 3 Report

great review

Author Response

(The authors gave the same response as above.)
